# *Bacillus subtilis* Response to Mercury Toxicity: A Defense Mediated by Sulphur-Rich Molecules and Oxidative Prevention Systems

**DOI:** 10.3390/ijms262010179

**Published:** 2025-10-20

**Authors:** Luis Fernando García-Ortega, Iliana Noemí Quiroz-Serrano, Jesús Guzmán-Moreno, Mario Pedraza-Reyes, Rosa María Ramírez-Santoyo, Luz Elena Vidales-Rodríguez

**Affiliations:** 1Departamento de Ingeniería Genética, Centro de Investigación y de Estudios Avanzados del Instituto Politécnico Nacional (CINVESTAV), Irapuato 36824, Mexico; luis.garcia@cinvestav.mx; 2Laboratorio de Biología de Bacterias y Hongos Filamentosos, Unidad Académica de Ciencias Biológicas, Universidad Autónoma de Zacatecas, Zacatecas 98066, Mexico; ilianaq1118@gmail.com (I.N.Q.-S.); jesumo@uaz.edu.mx (J.G.-M.);; 3Departamento de Biología, División de Ciencias Naturales y Exactas, Universidad de Guanajuato, Guanajuato 36050, Mexico; pedrama@ugto.mx

**Keywords:** mercury toxicity, *Bacillus subtilis*, mutagenesis, sulfur-rich molecules, oxidative stress, RNA seq

## Abstract

Upon reacting with cellular components, Hg(II) ions elicit the production of reactive oxygen species (ROS). While the ROS-promoted cytotoxic and genotoxic effects induced by Hg(II) have been widely described in eukaryotes, such effects have been less studied in bacteria. In this work, the prokaryotic environmental model *Bacillus subtilis* was employed to evaluate the cytotoxic and genotoxic impact of Hg(II) over strains proficient or deficient in SOS, general stress and antioxidant responses, as well as the global transcriptional response elicited by this ion. The exposure to HgCl_2_ significantly increased the mutation frequency to rifampicin resistance (Rif^R^) in WT and mutant strains, suggesting a major contribution of these pathways in counteracting the genotoxic effects of Hg(II). Detection of A → T and C → G transversion mutations in the *rpoB* gene of Hg(II)-exposed cells suggested the generation of 8-oxo-guanines (8-OxoGs) and other oxidized DNA bases. The RNA-seq study revealed upregulation of genes involved in efflux and/or reduction of metal ions, synthesis of sulfur-containing molecules, and downregulation of genes implicated in iron metabolism and cell envelope stress. Therefore, our results indicate that metal extrusion and scavenging of Hg(II) by thiol-rich molecules may constitute a line of defense of *B. subtilis* that counteracts the noxious effects of ROS resulting from an imbalance in iron metabolism elicited by this ion.

## 1. Introduction

Mercury (Hg) derived from industrial wastes is a toxic heavy metal that contaminates the environment and can have harmful effects on living organisms [1]. The presence of this element in the environment is caused by distinct anthropogenic activities, including the use of fossil fuels and pesticides, waste incineration, and mining, among others [2]. The most widely described mechanisms of cell damage in eukaryotes caused by mercury ions occur through oxidative stress, which is characterized by the overproduction of ROS [3]. The oxygen radicals, superoxide anion (O_2_^∙−^) and hydroxyl (OH^∙−^), as well as hydrogen peroxide (H_2_O_2_), can cause lipid peroxidation (which affects the membrane potential), protein and nucleic acids oxidation; the latter can promote mutagenesis or cytotoxic single and double DNA strand breaks [4,5]. The undesirable effects of free radicals can be prevented by enzymes like superoxide dismutase, catalase, and glutathione reductase, and non-enzymatic antioxidants such as ascorbate, glutathione, alkaloids, and tocopherols [4]. However, the high affinity of Hg(II) ions for the thiol groups (-SH) of reduced glutathione (GSH), superoxide dismutase (SOD), catalases (Kat) and glutathione peroxidase (GSH-Px) can interfere with or inactivate its catalytic properties [3,4].

In prokaryotes, a previous study in *Escherichia coli*, revealed that Hg(II) ions can bind to -SH groups of glutathione (GSH) and indistinctly to cysteine residues of proteins, causing a cellular homeostatic imbalance [6]. In *Corynebacterium glutamicum*, the exposure to mercury bi-chloride (HgCl_2_) increased the production of antioxidant enzymes such as NADPH-quinone reductases, zinc-dependent oxidoreductases, and thioredoxin reductase [7]. Although the induction of oxidative-damage prevention systems by Hg(II) ions has been documented in prokaryotic cells, the impact of Hg(II)-promoted oxidative DNA damage and its mutagenic potential has remained elusive. To counteract the mutagenic and cytotoxic effects derived from oxygen radical attack of DNA, bacteria deploy mechanisms of base incision/excision (BER) and mismatch (MMR) repair. Oxidative genetic lesions can not only promote mutations, but also the expression of DNA repair genes [8,9], and/or the activation of the SOS system at high levels of DNA damage, which promotes homologous recombination repair [10].

In *B. subtilis*, the mutagenic effects of toxic metal ions have been studied employing the frequency of mutations taking place in the *rpoB* gene which produce colonies with a rifampicin resistance (Rif^R^) phenotype [11,12]. It has been shown that the spectrum of mutations within *rpoB* can be influenced by the environment in which the microorganism is grown [13]. It has also been found that mutations in the *rpoB* gene of rifampicin-resistant *B. subtilis* clones (Rif^R^) are usually single nucleotide substitutions that result in specific amino acid changes located at two positions, Q469R or Q469K and H482Y. These substitutions occur within the subdomain of *rpoB* that corresponds to rifampicin resistance group I [14].

Evolution has equipped *B. subtilis* with an arsenal of adaptive responses to monitor and rapidly respond to adverse growing conditions, including nutrient limitation, sudden temperature changes, and exposure to distinct biologically toxic compounds. In contrast to *E. coli*, *B. subtilis* can proliferate in a wide diversity of habitats, making this bacterium an ideal model of study to unravel the cellular strategies deployed by bacteria to survive challenging environmental conditions [15,16,17]. Accordingly, the prokaryotic model *B. subtilis* was employed in this work to evaluate the mutagenic potential of Hg(II) ions, the global transcriptional response to Hg(II) exposition, as well as the role of prevention/repair systems in counteracting the genetic damage inflicted by mercury ions over environmental bacteria.

## 2. Results

### 2.1. Impact of Hg(II) on the Viability of Bacillus subtilis Strains Proficient and Deficient in Repair and Prevention Mechanism of ROS-Promoted DNA Damage

The effect produced by HgCl_2_ on the survival of the wild type and different mutant (Δ*katAkatB*, Δ*sodA*, Δ*recA*, Δ*sigB* and ΔGO) strains of *B. subtilis* 168 was analyzed. To this end, cultures of these strains propagated to the mid-exponential phase of growth were exposed to increasing doses of HgCl_2,_ and cell viability was determined as described in Materials and Methods. The mutant strains Δ*katA/*Δ*katB*, Δ*recA*, Δ*sigB*, and ΔGO, exhibited decreased LD_50_ and LD_90_ values after HgCl_2_ treatment in reference to the WT parental strain (Figure 1 and Appendix A). Therefore, these prevention/repair proteins seem to be involved in counteracting HgCl_2_ cytotoxicity. In contrast, the SodA-deficient strain exhibited similar levels of susceptibility to mercury treatment as those observed in the wild-type strain (Figure 1 and Appendix A).

### 2.2. Mutagenic Effect of Hg(II) in Vegetative Cells of Wild Type Bacillus subtilis 168 and Mutant Strains Deficient in Different Stress Responses

The mutagenic effect of HgCl_2_ in wild-type and mutant strains was determined from mutation frequencies to Rif^R^ in cultures exposed to HgCl_2_ and those that were not exposed (Figure 2). The mutation frequencies of all the strains tested significantly increased in the presence of HgCl_2_ with respect to the non-treated culture controls (Figure 2). The wild type, Δ*recA*, and Δ*katA* Δ*katB* strains showed the greatest increase in mutation frequency after HgCl_2_ exposure, whereas the mutation frequencies of Δ*sigB*, Δ*sodA*, and ΔGO strains increased to a lesser extent (Table 1).

### 2.3. Type of Mutations in Bacillus subtilis rpoB Induced by Hg(II) Ions

To investigate the types of Rif^R^ mutations associated with mercury ion treatment in *B. subtilis*, cluster I of the *rpoB* gene from 13 Rif^R^ colonies from WT *B. subtilis* cultures exposed to Hg(II) or left untreated (control) was subjected to DNA sequencing. Results revealed a similar frequency of the base transitions A → G in the position 1445 of *rpoB*; however, A → C transitions at position 1444 (H482D) were only detected in 3 Rif^R^ colonies of the Hg(II)-treated strain (Figure 3). Further transversion mutations in hotspots of the *rpoB*-cluster I were only detected in Hg-exposed Rif^R^ colonies; specifically, A → T transversion (1/13) at position 1406 (Q469L) (Figure 3).

### 2.4. Differentially Expressed Genes in Bacillus subtilis Exposed to Hg(II) Ions

Results from a transcriptomic analysis revealed that exposure of *B. subtilis* to Hg(II) ions resulted in upregulation of 28 genes and downregulation of 88 genes (Appendix A). Hg(II)-treatment induced the expression of genes involved in arsenite resistance (*arsB*, *arsC*, *arsF*, *aseR*), cysteine metabolism (*tcyB*, *tcyC*, *tcyJ*, *tcyK*, *tcyL*, *tcyM*, *tcyN*, *tcyP*), deoxyribonucleotide synthesis (*nrdI*, *nrdE*, *nrdF*, *ymaB*) putative membrane proteins (*yybF*, *yugS*, *ytnM*), membrane transport efflux proteins (*ydgK*, *copA*), biosynthesis of sulfur-containing cellular components (*yrkE*, *yrkF*, *yrkH*), biofilm formation (*tkmA*), phage protein (*yomN*), and genes encoding proteins with an unassigned function (*yvdD*, *ydhK*) (Table 2 and Appendix A). Gene ontology analyses revealed that upregulated genes were mostly grouped in molecular functions and/or biological processes related to membrane transport functions, L-cysteine transport, and deoxyribonucleotide synthesis mediated by ribonucleoside diphosphate reductase (Figure 4).

In contrast, downregulated genes resulting from Hg(II)-exposure were mainly associated with siderophore and iron metabolism such as uptake and synthesis of bacillibactin (*dhbABCE*, *besA*, *fosB*, *eesA*, *feuC*, *feuV*, *btr*, *ybfP*), iron assimilation and [Fe-S] cluster biosynthesis (*hmoA*, *sxzA*, *nifS*, *efeBM*, *frxB*, *ffoR*) as well as petrobactin and iron-compound uptake (*fhuCG*, *pbtOPQ*) (Table 3).

Additional downregulated genes included those related to tellurite (*yceCDEFGH*) and antibiotic (*ydbST*, *pbpE*, *yvdT*) resistance, phage proteins (*ydjGHIJ*, *liaH*, *liaI*, *yvlABCD*), protein metabolism (*yjoB*, *aeeB*), membrane lipid metabolism (*fldN*, *fldP*, *desE*, *floA*), metal ion transport (*ythQ*, *modA*, *modB*), starvation-induced killing (*skiWXYZ*), alkyl hydroperoxide reductase (*ahpC*), membrane function and transport (*yvaCDEF*, *yteJ*, *psmAB*, *csbA*, *msmX*). Notably, most of these downregulated genes belongs to the sigmaW (SigW) regulon, which is known to be activated by sigmaW factor in response to cell envelope stress (Table 4).

Additional genes with a downregulation status included *yfhC* encoding a putative oxidoreductase (nitroreductase family), *spo0M* (protein involved in the control of cell cycle as a function of the environment), *bscR* (transcriptional regulator for *cypB*), *pftB* (pyruvate import system subunit B), *pftA* (pyruvate uptake system subunit A), *sppA* (signal peptide peptidase), *padC* (phenolic acid decarboxylase), *yvdQ* (conserved protein of unknown function), *yxlE* (negative regulator of sigma-Y activity), and *yxzE* (putative bacteriocin). Diverse putative proteins, including *ydcC* (putative lipoprotein), *ydgG* (putative transcriptional regulation (MarR family)), *ydgH* (putative membrane component), *ydjP* (putative aminoacrylate hydrolase), conserved proteins of unknown function (*yzzP*, *yocL*, *yqfB*, *yxjI*, *yvdQ*), and hypothetical proteins (*yeaA* and *ykuO*), were also downregulated. Some of the proteins encoded by these genes also belong to sigmaW-regulon (Table 4 and Appendix A).

In general, mostly downregulated genes were grouped in biological processes such as membrane functions, homeostasis, and iron metabolism (Figure 4).

## 3. Discussion

The aim of this work was to evaluate the mutagenic potential of Hg(II) ions and the bacterial stress responses that can counteract its toxicity. To this end, *B. subtilis* was used as a model, and mutagenesis caused by Hg(II) ions was assessed by determining mutation frequencies to rifampicin of wild-type *B. subtilis* 168 and strains deficient in KatA, KatB, RecA, SigB, and the GO system in order to determine whether antioxidative or general stress responses are implicated in counteracting the Hg-induced damage. Additionally, RNAseq analysis was carried out to identify genes and/or general pathways that counteract mercury toxicity. Overall, our results support that Hg(II) ions can induce mutagenesis in *B. subtilis* by oxidative damage to DNA and the primary defense response against mercury toxicity relies on scavenge thiol-rich molecules to avoid the imbalance of iron metabolism and loss of homeostasis.

The negative impact on cell viability observed in *B. subtilis* cells exposed to Hg(II) ions confirms the toxicity of this heavy metal. Moreover, the increased susceptibility to Hg(II) ions exhibited by cells deficient in KatA, KatB and the GO system supports the notion that toxicity is derived from ROS-promoted DNA damage, as was previously postulated [4,5]. Notably, the lack of SodA, an antioxidant enzyme that converts superoxide radicals into hydrogen peroxide [18], did not show differences in the resistance to Hg(II) ions compared to resistance observed in the wild-type strain. In agreement with this observation, it has been reported that the SodA-deficiency did not increase the susceptibility of *B. subtilis* cells to Cr(VI) [11]. Therefore, it is feasible that in the absence of SodA, additional systems confer protection from oxidative stress promoted by heavy metals to this bacterium. Indeed, in low (G+C)-content Gram-positive bacteria like *Bacillus*, bacillithiol (BSH), a LMW thiol [19], could efficiently protect the cells against superoxide and metal stress in the absence of SodA [20].

The induction of oxidative-damage prevention systems by Hg(II) ions has been documented in eukaryotic [21] and prokaryotic organisms [6,7]; however, the impact of oxidative stress promoted by Hg(II) ions on DNA and its mutagenic potential remain elusive. It has been shown that prokaryotes rely on distinct DNA repair pathways to cope with the harmful effects produced by the increasing levels of ROS [8,9,10].

In this work, it was found that exposure to Hg(II) increased the mutation frequency to Rif^R^ of wild-type *B subtilis* cells~2.6 times, thus attesting to the mutagenic potential of this heavy metal in bacteria. It is possible that ROS-promoted base oxidation due to the accumulation of 8-OxoG and derived repair intermediates is involved in this process; in support of this notion, as shown in this work, is the fact that the lack of a functional GO system increased the mutagenesis in this microorganism (Figure 2). Furthermore, increased levels of the RNR operon (*nrdI*-*nrdE*-*nrdF*) were detected in *B. subtilis* cells amended with Hg (II), suggesting a higher requirement of deoxyribonucleotides (dNTPs) for DNA repair processes. Indeed, the repression of RNR operon transcription leading to unbalances in dNTPs pools has been found to promote stress-associated mutagenic processes in *B. subtilis* [22].

However, the mutation frequencies determined in *B. subtilis* strains deficient in RecA and the GO system, amended with Hg(II), increased in similar levels to those observed in the wild-type strain (Table 1), suggesting that additional antioxidant determinants in addition to catalases, superoxide dismutase, RecA, and the GO system, prevent the ROS-promoted mutagenic effects of mercury ions in *B. subtilis*.

The molecular analyses of *rpoB* leading to identification of base substitutions associated with the Rif^R^ phenotype elicited by mercury ions revealed predominance of C → T and A → G transitions resulting from adenine and cytosine deamination at positions 1444 and 1445 of *rpoB*, respectively [23]. Strikingly, 3 out of 26 Rif^R^ colonies analyzed showed the *rpoB* G → C transition; however, such mutation was exclusively detected in Rif^R^ colonies derived from Hg(II) treatment. Therefore, this mutation derived from missreplication of the oxidative lesion 5-OxoG [23,24] can be considered as a landmark of Hg(II)-promoted mutagenesis in *B. subtilis*. Overall, the mutational spectrum promoted by Hg(II) in this bacterium can be associated with oxidation of nucleobases as even base deamination has been found to be exacerbated by oxidative stress and this type of damage can be mainly processed through the BER pathway [25,26].

The global transcriptional response of *B. subtilis* cells exposed to Hg(II) ions suggest that the first line of response occurs through the scavenging of thiol-rich molecules that function as nonenzymatic antioxidants that interact with Hg(II) ions to prevent oxidative damage. This notion is supported by the observed induction of genes involved in metabolism of thiol-rich molecules, including the transport of cystine, biosynthesis of sulfur-containing cellular components and transport of sulfur-containing amino acids (Table 2). Notably, a proteomic study in *B. subtilis* cells deficient in the GO system, prone to accumulate 8-OxoG lesions, paralleled these results [27], supporting the notion of a ROS-promoted mechanism of genotoxicity elicited by Hg(II) in this microorganism.

Additionally, overexpression of metal membrane transport efflux-associated genes, such as *arsB*, *arsF*, and *copA*, suggest that it could be useful for extrusion of Hg(II) ions. In this regard, *B. subtilis* possesses five ArsR/SmtB paralogues, seven MerR homologues, and seven putative metal ion efflux proteins that contribute to general metal ion homeostasis [28]. In particular, the expression of *arsB* (encoding an arsenite efflux pump) and *arsF* (encoding an arsenite/antimonite/H^+^ antiporter) is commonly regulated by an ArsR family repressor that respond to As(III) and Sb(III) [29] but also is strongly induced by the divalent cation Cd(II) and the monovalent cation Ag(I) [30].

In this study, *aseR* (encoding a paralogue of ArsR) was found to be induced by Hg (II), while a previous transcriptional study reported a strong induction of this gene as part of the earliest transcriptional response of *B. subtilis* cells by stress-inducing concentrations of Ag(I), As(V), As(III), Cd(II), Co(II), Ni(II), Zn(II), and Cu [added as Cu(II)] [30]. To our knowledge, our study reports for the first time the transcriptional induction of *aseR* by Hg(II). In a similar manner, the expression of *copA* (P-type ATPase dedicated to Cu efflux) in *B. subtilis* is positively regulated by CueR (a MerR homolog) in response to elevated Cu levels [31]; however, CueR also responded strongly to other monovalent cations as Ag(I) and divalent ions as Cd(II), and weakly to Zn(II), suggesting that CueR is able to detect divalent as well as monovalent ions [30]. Thus, the expression of *copA* in response to Hg(II) ions observed in this study could be due to a positive regulation by CueR in response to Hg(II) ions. Moreover, a study corroborates that genes most strongly induced by sudden exposure to metal ions include arsenate/arsenite resistance proteins (encoded by the *ars* operon), the CadA and CopA P-type efflux ATPases, and the cation diffusion facilitator (CDF) protein, CzcD [30]. Particularly, ArsR (a transcriptional regulator that derepress arsenical stress response genes of the *ars* operon in response to As(III) and Bi(III)) [29] was also strongly induced by the divalent cation Cd(II) and the monovalent cation Ag(I) [30].

The transcriptional profiles of *B. subtilis* cells exposed to Hg(II) also revealed downregulation of genes of iron metabolism as those involved in the uptake and synthesis of siderophores (bacillibactin and petrobactin), iron-compound uptake, iron assimilation and [Fe-S] cluster biosynthesis. These results correlated with reports indicating that in *E. coli* cellular processes such as iron homeostasis and electrolyte balance are disrupted as a consequence of mercurial compounds interacting with cysteine sites of several hundreds of proteins and the formation of stable adducts [32]. Moreover, although there is no evidence that mercury itself undergoes Fenton-type chemistry to generate reactive oxygen species like iron and copper [33], it was reported that Ag(II) and Hg(II) ions can degrade [Fe-S] clusters, promoting the accumulation of unincorporated Fe(II) ions in cellular pools, thus accelerating Fenton reactions with the subsequent oxidation of biomolecules [34,35,36]. Thus, downregulation of genes related to uptake and assimilation of iron may be due to the accumulation of Fe(II) ions in cellular pools; the high concentrations of Fe(II) promote the iron-dependent repression of genes (mediated by the metalloregulatory repressor Fur) involved in the high affinity uptake pathways for iron compounds [37]. The accumulation of Fe(II) in cellular pools may also explain the downregulation of *ahpC* (one of the subunits of the AhpC/AhpF alkyl hydroperoxide reductase (AHPR)), which is regulated by PerR (a Fur homolog) that represses a set of genes (including *ahpCF*) involved in scavenging the mutagenic ROS H_2_O_2_ and organic peroxides. Increased intracellular iron levels [37] activate PerR (a repressor activated by Fe(II) binding), thereby enhancing its repressor function and leading to *ahpCF* repression. Notably, repression of AhpC synthesis has also been associated with the mutagenic phenotype of a *B. subtilis* strain deficient in the GO system for prevention/repair and prone to accumulating 8-OxoG lesions [27].

The downregulation of genes of the cell envelope stress response was also elicited by Hg(II); interestingly, several of these genes are part the of extracytoplasmic function (ECF) regulon, coordinated by SigW, a RNApol sigma factor (σ^w^) [38,39,40]. The genes that belong to this regulon have a known or predicted role in detoxification, defense against phage infection or antibiotics, synthesis of bacteriocins, alkaline shock response, modulation of lipid composition, and membrane protection and remodeling [38,41]. In general, the cell envelope stress response in *B. subtilis* is mediated by SigW functions through membrane proteins such as FloA and FloT (flotillin homologs), the two members of the phage shock protein family (LiaH, PspA), as well as genes related to antibiotic-specific detoxification modules (i.e., *fosB*, *pbpE*, *yvdTQ*, *yxzE*, *ydbST*, *ydjP*, *yceE*) [38,39,41,42]. The downregulation of a set of SigW-dependent genes observed in *B. subtilis* exposed to Hg(II) (Table 4), including those related to cell envelope functions such as the modulation of membrane fluidity (*floA*, *desE*, *fldNP*), phage shock (*liaHI*, *pspA*, *ydjGHIJP*, *yvlABCD*), detoxification and antibiotic protection (*yxzE*, *pbpE*, *yvdTQ*, *yceCDEFGH*, *ydbST*, *skiWXYZ*), and membrane transport (*yteJ*) suggest that Hg(II) ions do not promote envelope cellular stress. This contention is in agreement with studies reporting that in the absence of the stresses that activates the SigW-regulon, SigW is downregulated by the anti-sigma factor RsiW, which is localized in the plasma membrane [43]. This is also consistent with the phenotype observed in a *sigW* mutant of *B. subtilis* on which the downregulation of genes activated by SigW was observed [39]. An alternative explanation for downregulation of the SigW-regulon genes is that Hg(II) ions could inhibit the SigW-mediated response by alteration of the proteolytic functions of the transmembrane proteins PrsW and RasP, two metalloproteases responsible for proteolysis of the anti-sigma factor RsiW, a membrane protein that inactivates SigW through a tight interaction and prevents transcriptional activation of genes in this regulon [44,45,46]. This hypothesis is supported by the potential of Hg(II) ions to the replacement of metal cofactors with essential function in proteins, particularly iron and zinc metalloproteins [33,47,48]. However, further studies are required to support this.

Based on experimental evidence obtained in this study, we postulate a mechanism of cell toxicity and stress response elicited by mercury in *B. subtilis* (Figure 5). The observed downregulation of genes involved in the uptake and assimilation of iron, as well as in [Fe-S] cluster biosynthesis, strongly suggests that upon interacting with thiol groups of cysteine-rich proteins, Hg ions can remove Fe(II) atoms from proteins containing [Fe-S] cluster [33,35,36,37]. These processes can increase the iron pools and propel the Fenton reaction, thus eliciting ROS-promoted cell damage (i.e., mutagenesis). Furthermore, the downregulation of genes belonging to the SigW-regulon observed in our study also suggests a deficient envelope cell stress response dependent on SigW [38,39,40]. Therefore, as a first line of defense against Hg(II) toxicity, *B. subtilis* seems to employ the synthesis of cysteine- and thiol-rich molecules for targeting mercuric ions, which exhibit a strong affinity for reduced sulfur atoms (Figure 5, left side) [49]. This mechanism may prevent *B. subtilis* from the attack of iron–sulfur clusters by Hg(II) and the consequent induction of ROS-promoted DNA damage described above.

Finally, we propose that the toxic effects of Hg ions may also result from increases in free cellular iron concentration negatively impacting the transcriptional functions of the repressors Fur and PerR (Figure 5, right side). Accordingly, Hg treatment repressed the expression of genes involved in uptake transport and iron metabolism, as well as *ahpC* encoding an alkyl hydroperoxide reductase that protects *B. subtilis* from the noxious effects of organic peroxides [37].

## 4. Materials and Methods

### 4.1. Bacterial Strains, Culture Conditions, and Reagents

All the bacterial strains used in this study are derived from the *B. subtilis* 168 strain (Table 5). The strains were grown in LB medium (Luria Broth) supplemented with Neomycin (Neo; 10 µg/mL), Tetracycline (Tet; 10 µg/mL), Chloramphenicol (Cm; 5 µg/mL), Erythromycin (Ery; 5 µg/mL), or Rifampicin (Rif; 10 µg/mL) as required. Liquid cultures were grown in antibiotic A3 medium and incubated with shaking at 200 rpm at 37 °C. Solid medium cultures were grown at 37 °C. For the evaluation of mercury toxicity, a 200 µM stock solution of mercury dichloride (HgCl_2_) was used to supplement the culture with Hg(II) ions at different concentrations.

### 4.2. Effect of Hg(II) Ions on Growth of Bacillus subtilis and Mutant Strains Deficient in Genes Related to Different Stress Responses

For determining Hg(II) toxicity, liquid cultures of wild-type and mutant strains (Table 5) were propagated to the midpoint of the exponential growth phase. An amount of 50 mL of each strain culture was transferred to 125 mL Erlenmeyer flasks, exposed to increasing doses of HgCl_2_ (0, 0.5, 1, 1.5 and 2 µM), and then incubated during 2 h at 37 °C and 200 rpm. To evaluate the effect of HgCl_2_ on the survival of *B. subtilis*, the bacterial cell viability of Hg(II)-exposed cultures was determined by the serial dilution method. Bacterial viability data, expressed as Colony Forming Units per mL (CFU/mL), was plotted to obtain a dose–response curve used to determine the lethal doses 50 (LD_50_) and 90 (LD_90_) of HgCl_2_.

### 4.3. Analysis of Mutagenesis Induced by Hg(II) Ions

The mutagenic effect of Hg(II) ions on *B. subtilis* cells was determined by the frequency of rifampin resistance (Rif^R^) colonies produced in cells exposed to HgCl_2_ and those left untreated (control). Briefly, 50 mL of a liquid culture of *B. subtilis* at mid exponential growth phase was divided into two subcultures; one of them was treated with a LD_50_ of HgCl_2_ and the other was untreated. Both cultures were incubated for 2 h at 37 °C and 200 rpm. Afterwards, aliquots of 0.1 mL of each culture were plated on six LB agar plates supplemented with rifampicin (10 µg/mL) and incubated at 37 °C during 24 h. The number of colonies resistant to rifampicin (Rif^R^) was determined and the cellular viability in both exposed and unexposed cultures was determined by the serial dilution method. The mutation frequency to rifampicin was calculated based on the number of Rif^R^ mutants with respect to the number of viable cells in the cultures.

### 4.4. Analysis of Mutations Induced by Hg(II) Ions in Bacillus subtilis

The identification of mutations in the *rpoB* gene of the Rif^R^ mutants obtained after exposure to Hg(II) ions was carried out as follows. Twenty-six Rif^R^ colonies were isolated from the mutation frequency experiments (13 Rif^R^ colonies from the control untreated culture and 13 Rif^R^ colonies from HgCl_2_ treated cultures) and the chromosomal DNA was extracted. The isolated DNA was used as template for the amplification of a 708 pb fragment of the *rpoB* gene (nt +1353 to +2061 in relation to the open reading frame of the *rpoB* gene), this fragment contains the three groups of the hot spots where mutations that confer resistance to rifampicin occur in many bacteria, including *B. subtilis* [51]. The PCR reaction was performed using High Fidelity Platinum *Pfx* DNA Polymerase (Invitrogen, Carlsbad, CA, USA) (1U) and the oligonucleotide set: direct 5′-CGT CCT GTT ATT GCG TCC-3′ and reverse 5′-GGC TTC TAC GCG TTC AAC G-3′. The amplified products were sequenced, and the sequences obtained were analyzed for detection of the specific mutations which confer rifampicin resistance.

### 4.5. RNA Seq Analysis

#### 4.5.1. RNA Isolation, Data Processing, and Sequencing

For the RNA isolation, two liquid cultures of wild-type *B. subtilis* 168 at midexponential growth phase were prepared; one of them was exposed to an LD_50_ of HgCl_2,_ and the other was unexposed and used as a control. Both cultures were incubated for 2 h at 37 °C and 200 rpm. Aliquots of 1 mL were centrifuged at 12,000 rpm for 1 min. The cell pellet was used for RNA extraction using the Quick-RNA^TM^ Fungal/Bacterial Miniprep kit (ZYMO RESEARCH, Irvine, CA, USA). RNA samples were spectrophotometrically quantified in NanoDrop^TM^ (Thermo Scientific, Wilmington, DE, USA) and RNA quality determined by electrophoresis in 1% agarose gel [52]. For RNAseq, the quality analysis and quantification of RNA samples was carried by fluorimetry; cDNA libraries were obtained and the amplicons sequenced by Illumina technology (NOVOGENE, Sacramento, CA, USA). 

#### 4.5.2. Transcriptome Analysis

Raw reads, NCBI’s Sequence Read Archive (SRA) with access ID PRJNA1304564, were first cleaned with fastp v0.23.4 [53] to remove reads of low quality, potential adaptor sequences, and long terminal homopolymeric stretches. Clean reads were then aligned and quantified using kallisto v0.48.0 [54] against the cDNA transcript sequences reported for strain 168 (ASM904v1) in the NCBI database. To identify anomalous samples, an outlier map was made based on the robust score distances and orthogonal distances computed by the R function PCAGrid v1.7–5 [55] for the normalized abundance matrix as suggested by Chen et al. (2020) [56]. Based on the results, one replicate of control and one for Hg(II) treatment were removed.

#### 4.5.3. Differential Expression and Functional Analyses

To identify differentially expressed genes, the exactTest function of the edgeR v4.2.2 package [57] was used. The resulting *p*-values were adjusted using the *q*-value function, estimating the false discovery rate (FDR). Genes with a *q*-value ≤ 0.01 were considered differentially expressed, regardless of the fold-change. Gene Ontology enrichment analysis of the differentially expressed genes was performed using the PANTHER v19.0 platform [58] with Fisher’s exact test and FDR correction. Only categories with FDR < 0.05 were considered significant. To summarize the enriched GO terms, we utilized the REVIGO v1.8.2 online analysis tool [59], allowing a similarity score of 0.7 between GO terms.

### 4.6. Statistical Analysis

The bacterial growth and dose–response curves were performed at least three times, and data were represented as mean ± Standard Deviation (SD). LD_50_ and LD_90_ values were calculated from the linear regression equation obtained from three independent dose–response curves. The data was analyzed using Student’s *t*-test to determine differences in mutants with respect to the WT strain with *p* value < 0.05 (*p* < 0.05). The Rif^R^ mutation frequency was performed at least three times and data were represented as mean ± Standard Deviation (SD), since the data presented a normal distribution (using the test Shapiro–Wilk). The parametric Student’s *t*-test (*p* < 0.05) was used to determine statistical differences in the mutation frequency between Hg-exposed and non-exposed parental and mutant strains. The data were analyzed using past program version 4.03 for Windows 10.

## 5. Conclusions

In this work, we employed *Bacillus subtilis* as a Gram-positive model to investigate the cytotoxicity, genotoxicity, and global transcriptional response elicited by the heavy metal pollutant mercury in environmental bacteria. In summary, our results indicate that upon interacting with cysteine-rich molecules, Hg(II) promote iron and cysteine metabolism stresses. Such processes increase the transcription of genes encoding cysteine-rich molecules to counteract mercury toxicity. High intracellular iron concentrations generate ROS-promoted DNA damage. In support of these notions, antioxidant and BER proteins were found to prevent the cytotoxicity and genotoxicity of Hg(II). However, the mechanistic aspects by which DNA mutations are induced by mercury in *B. subtilis*, represent an important topic for future research.

## Figures and Tables

**Figure 1 ijms-26-10179-f001:**
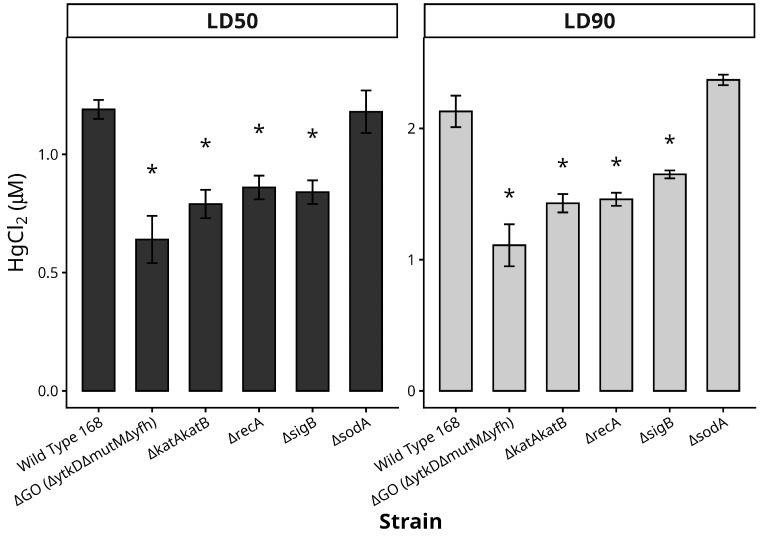
Lethal doses of HgCl_2_ in wild type *Bacillus subtilis* 168 and mutant strains deficient in different oxidative stress responses. The bars show the main lethal doses of HgCl_2_ for the wild-type 168 and the mutant strains. Asterisks (*) indicate a statistically significant increase in sensitivity to HgCl_2_ (Student’s *t*-test, *p* < 0.05) compared to the wild-type strain.

**Figure 2 ijms-26-10179-f002:**
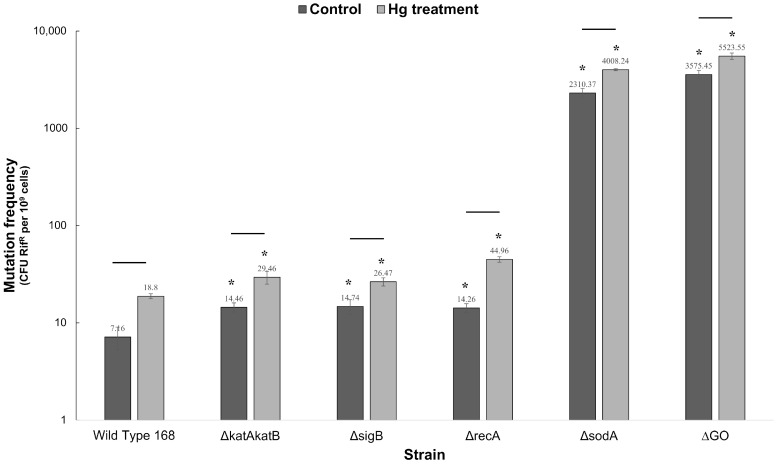
Mutagenic effect of HgCl_2_ in the different *B. subtilis* strains. The strains, wild-type 168, Δ*katAkatB*, Δ*sigB*, Δ*recA*, Δ*sodA*, and ΔGO, were grown in A3 medium until reaching the mid exponential growth phase at OD_600nm_ and then they were divided into 2 equal cultures; one of them was treated with a LD_50_ of HgCl_2_ and the other was used as a control (untreated). The Rif^R^ frequency mutation was determined as described in materials and methods. The asterisks indicate statistical differences (Student’s *t*-test with *p* value < 0.05) with respect to the WT strain. Lines above the bars indicate statistical difference between untreated cultures and cultures treated with HgCl_2_ (Student’s *t*-test with *p* value < 0.05).

**Figure 3 ijms-26-10179-f003:**
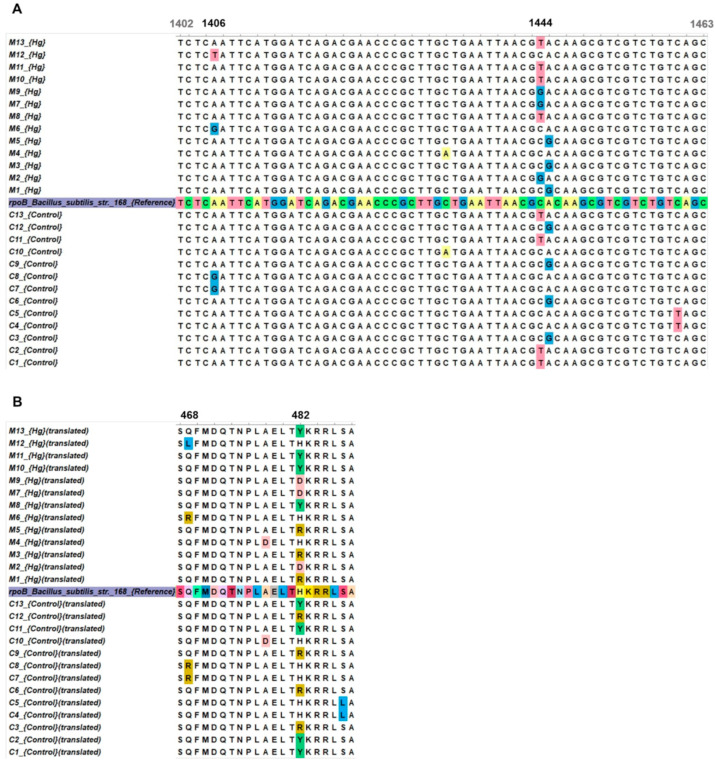
Mutations in cluster I of the *B. subtilis rpoB* gene induced by Hg(II) ions. Multiple alignment of (**A**) nucleotide and (**B**) amino acid sequences. The *B. subtilis* 168 *rpoB* reference sequence is shown in the middle and highlighted in color. C1–C13 are control sequences, and M1–M13 are sequences from clones exposed to mercury. Nucleotides and amino acids that changed relative to the control are shown in color. Numbers in bold indicate the positions of nucleotides and amino acids that changed due to mercury exposure.

**Figure 4 ijms-26-10179-f004:**
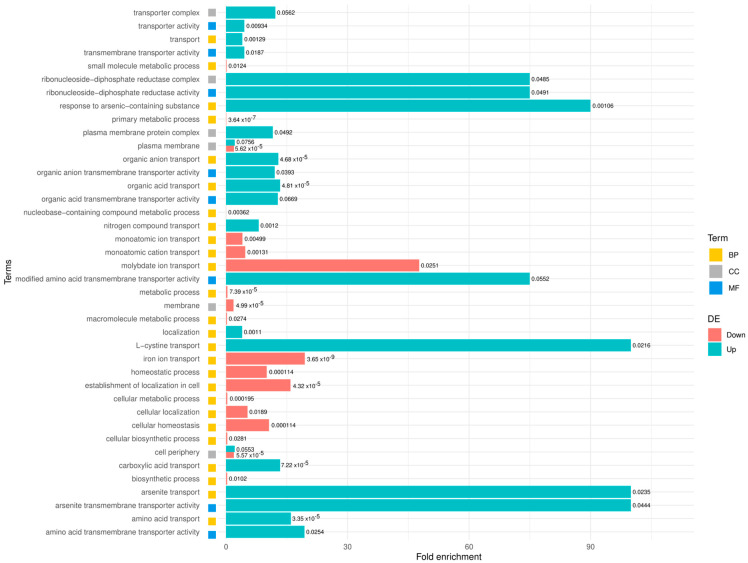
Gene Ontology (GO) enrichment analysis results for deferentially expressed genes according to expression profile. BP = biological process, CC = cellular component, and MF = molecular function. Coral bars indicate down-regulated genes; cyan bars indicate up-regulated genes.

**Figure 5 ijms-26-10179-f005:**
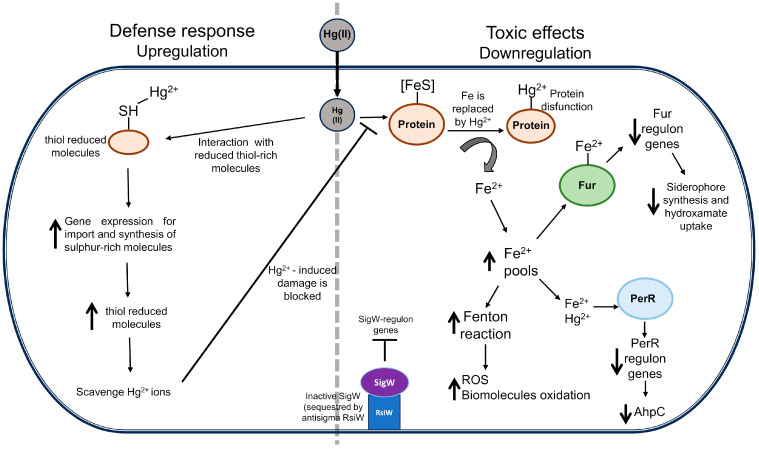
Proposed model of the toxic effects of Hg(II) ions and cellular response to mercury-induced damage in *B. subtilis*. Right side shows the toxic effects induced by exposure to Hg(II) ions. Iron from [FeS]-proteins is replaced and released by Hg(II) ions. Intracellular free Fe(II) ions increase and accelerate the Fenton reaction with the consequent induction of oxidative damage to biomolecules and repression of genes regulated by Fur and PerR regulons. The SigW-regulon genes remain repressed by the absence of envelope stress, on which SigW is sequestered by the anti-sigma RsiW. Left side shows the defense response induced by Hg(II) ions. Hg(II) ions are scavenged by reduced thiol-rich molecules producing a decrease in the reduced thiol-molecules and the consequent gene expression for the import and synthesis of reduced sulfur-rich molecules necessary to block the oxidative damage induced by Hg(II) ions. (↑) increase, (↓) decrease.

**Table 1 ijms-26-10179-t001:** Mutation frequency in *Bacillus subtilis* strains exposed and unexposed to Hg(II) ions.

Strain	Mean Rif^R^ Mutation Frequency ± SD (Rif^R^ Per 10^9^ Cells) − HgCl_2_	Mean Rif^R^ Mutation Frequency ± SD (Rif^R^ Per 10^9^ Cells) + HgCl_2_	Relative Frequency
WT 168	7.16 ± 1.52	18.8 ± 0.89	2.62
Δ*recA*	14.26 ± 1.46	44.96 ± 2.87	3.15
Δ*sigB*	14.74 ± 2.58	26.47 ± 2.55	1.79
Δ*katAkatB*	14.46 ± 1.56	29.46 ± 4.42	2.03
Δ*sodA*	2310.37 ± 250.93	4008.2 ± 98.08	1.73
Δ*GO* (Δ*ytkD*Δ*mutM*Δ*yfhQ*)	3575.45 ± 355.144	5523.55 ± 426.86	1.54

**Table 2 ijms-26-10179-t002:** Upregulated cellular functions of *Bacillus subtilis* in response to the Hg(II) ions exposure.

Gene	Description	Function
*tcyC*	Cystine ABC transporter (ATP-binding protein)	Cysteine metabolism
*tcyB*	Cystine ABC transporter (permease)	
*tcyP*	(Sodium)-cystine symporter	
*tcyN*	Sulfur-containing amino-acid ABC transporter (ATP-binding protein)	
*tcyM*	Sulfur-containing amino acid ABC transporter (permease)	
*tcyL*	Sulfur-containing amino acid ABC transporter (permease)	
*tcyK*	Sulfur-containing amino acid ABC transporter binding lipoprotein	
*tcyJ*	Sulfur containing amino acid ABC transporter binding lipoprotein	
*nrdI*	Cofactor of ribonucleotide diphosphate reductase	Deoxyribonucleotide synthesis
*nrdE*	Ribonucleoside-diphosphate reductase (major subunit)	
*nrdF*	Ribonucleoside-diphosphate reductase (minor subunit)	
*ymaB*	Putative cofactor involved in deoxyribonucleotide synthesis	
*ydgK*	Putative efflux transporter	Membrane transport efflux
*copA*	Copper transporter ATPase	
*arsC*	Thioredoxin-coupled arsenate reductase; skin element	Arsenic resistance
*arsB*	Arsenite efflux transporter; skin element	Arsenic resistance/transport efflux
*yrkH*	Putative sulfur transferase/hydrolase	Biosynthesis of sulfur-containing compounds
*yrkF*	Putative rhodanese-related sulfur transferase	
*yrkE*	Putative protein involved in sulfur metabolism (DsrE-like)	
*ytnM*	Putative transporter	Utilization of alternative sulfur sources

**Table 3 ijms-26-10179-t003:** Downregulated cellular functions of *Bacillus subtilis* in response to the Hg(II) ions exposure.

Gene	Description	Function
*feuV*	Iron(III)-siderophore transporter (ATP binding component)	Iron metabolism
*feuC*	Iron-uptake protein	
*ffoR*	Fur-regulated NADPH: ferredoxin oxidoreductase	
*hmoA*	Heme-degrading monooxygenase	
*sxzA*	Xenosiderophore schizokinen (dihydroxamate) transporter (permease)	
*nifS*	Desulfurase involved in iron-sulfur clusters for NAD biosynthesis	
*efeB*	Peroxidase converting ferric iron into ferrous iron	
*efeM*	Lipoprotein binding ferrous or ferric iron for transport	
*frxB*	Desferrioxamine-and ferrichrome-binding transporter lipoprotein (shuttle system)	
*pbtO*	Petrobactin iron-siderophore ABC transporter (permease)	Petrobactin transport
*pbtP*	Petrobactin iron-siderophore ABC transporter (ATP-binding protein)	Petrobactin uptake
*pbtQ*	Petrobactin iron-siderophore ABC transporter (binding lipoprotein)	
*fhuC*	Ferrichrome ABC transporter (ATP-binding protein)	
*fhuG*	Ferrichrome ABC transporter (permease)	Iron compound uptake
*yvaC*	Putative integral inner membrane protein	Small multidrug resistance
*yvaD*	Putative integral inner membrane protein	
*yvaE*	Putative metabolite-efflux transporter	
*yvaF*	Putative transcriptional regulator	
*yjoB*	Informational ATPase possibly involved in protein degradation	Protein metabolism
*aeeB*	L-Ala-D/L-Glu epimerase	
*fldN*	Short-chain flavodoxin (acts in lipid desaturation)	Membrane lipid metabolism
*fldP*	Short-chain flavodoxin	
*desE*	Fatty acid desaturase	
*psmA*	Sodium/proton antiporter subunit A	Membrane transport
*psmB*	Sodium-proton two component antiporter subunit	
*dhbB*	Isochorismatase (siderophore specific)	Bacillibactin synthesis
*dhbE*	2,3-dihydroxybenzoate-AMP ligase	
*dhbC*	Isochorismate synthase (siderophore-specific)	
*dhbA*	2,3-dihydro-2,3-dihydroxybenzoate dehydrogenase	
*besA*	Bacillibactin trilactone hydrolase	
*eesA*	Iron-chelator (enterobactin family) esterase	
*btr*	Transcriptional activator (AraC/XylS family) of synthesis and uptake of the siderophore bacillibactin	
*ahpC*	Alkyl hydroperoxide reductase (small subunit)	Response to exogenous H_2_O_2_-induced stress

**Table 4 ijms-26-10179-t004:** Downregulation of sigmaW-regulon genes in response to Hg(II) ion exposure.

Gene	Description	Function
*yceC*	Putative stress adaptation protein (tellurite resistance)	Tellurite resistance and response to ethanol-induced stress
*yceD*	Putative stress adaptation protein (tellurite resistance)	
*yceE*	Putative stress adaptation protein (tellurite resistance)	
*yceF*	Putative stress adaptation transporter (tellurite resistance)	
*yceG*	Putative toxic compound adaptation protein (tellurite resistance)	
*yceH*	Putative reactive oxygen species resistance protein	
*ydbS*	Resistance to heterologous antibiotics	Response to antibiotic-induced stress
*ydbT*	Resistance to heterologous antibiotics	
*fosB*	Magnesium-dependent bacillithiol-transferase (fosfomicin resistance)	
*liaH*	Modulator of *liaIHGFSR* (*yvqIHGFEC*) operon expression	
*liaI*	Membrane anchor for the phage-shock protein A homolog LiaH	
*pbpE*	Penicillin-binding protein 4	
*yvdT*	Putative transcriptional regulator (TetR/AcrR family)	
*yvdQ*	Conserved protein of unknown function	
*pspA*	Phage shock protein A homolog regulator; prophage region 3	Response to phage-induced stress
*ydjG*	Putative phage replication protein; prophage region 3	
*ydjH*	Conserved hypothetical protein; prophage region 3	
*ydjI*	Putative phage protein	
*ydjJ*	Putative membrane associated potassium channel; prophage region 3	
*ydjP*	Putative aminoacrylate hydrolase	
*yvlD*	Putative integral phage holin-like membrane protein	
*yvlC*	Membrane associated phage-like stress regulator, nisin resistance	
*yvlB*	Conserved protein of unknown function, stress-related	
*yvlA*	Conserved protein of unknown function	
*yeaA*	Conserved hypothetical protein	Response to cell envelope stress
*spo0M*	Protein involved in the control of the cell cycle as a function of the environment	
*floA*	Flotillin-like protein involved in membrane lipid rafts	
*skiW*	Subunit of permease exporting the starvation-induced killing protein	Starvation induced killing
*skiX*	Subunit of efflux permease exporting the starvation-induced killing protein	
*skiY*	Subunit of efflux permease exporting the starvation-induced killing protein (ATP-binding protein)	
*skiZ*	Permease subunit exporting Sporulation-Delaying Protein	
*yteJ*	Putative integral inner membrane protein	Membrane transport
*ythQ*	Putative ABC transporter (permease)	

**Table 5 ijms-26-10179-t005:** Genotypes of bacterial strains used in this study.

Strain	Genotype	Mutated Gene(s) and Function	Source/Reference
*B. subtilis*			
WT 168 PERM311	*trpC2*	Non-mutant parental strain	Lab. Stock Dr. Mario Pedraza Reyes
PERM741 (Δ*recA*)	*trpC2*Δ*recA::cat* Cm^R^	RecA protein, controls the SOS-transcriptional response which is triggered by DNA-damaging agents	Lab. Stock Dr. Mario Pedraza Reyes
PERM342 (Δ*sigB*)	*trpC2*Δ*sigB::cat* Cm^R^	Sigma B transcriptional factor, controls the general stress regulon	Lab. Stock Dr. Mario Pedraza Reyes
PERM1245 (∆*katAkatB*)	*trpC2*Δ*katAkatB::cat* Cm^R^ Eri^R^	KatA and KatB catalases, antioxidant enzymes that catalyze decomposition of hydrogen peroxide (H_2_O_2_) into H_2_O and oxygen (O_2_)	Lab. Stock Dr. Mario Pedraza Reyes
PERM434 (∆*sodA*)	*trpC2*Δ*sodA::cat* Cm^R^	SodA superoxide dismutase, antioxidant enzyme which catalyzes dismutation of superoxide (O^−2^) anion radical into molecular oxygen (O_2_) and hydrogen peroxide (H_2_O_2_).	[50] (Casillas-Martínez & Setlow, 1997)
PERM1136 (∆*ytkD∆mutM*∆*yfhQ*)	*trpC2* ∆*ytkD::neo* ∆*mutM::tet* ∆*yfhQ::eri* Neo^R^Tc^R^Eri^R^	DNA glycosylases MutM and YfhQ (*E. coli* MutY homologus), components of the base excision repair pathway. 8-oxo-dGTPase YtkD (*E. coli* MutT homologous) hydrolyzes triphosphate 8-oxo-dGTP. These three proteins are referred to as the oxidized guanine (GO) system	Lab. Stock Dr. Mario Pedraza Reyes

## Data Availability

The original data presented in the study are openly available in NCBI BioProject with access ID PRJNA1304564.

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
