# Peer review of "Bacillus subtilis Response to Mercury Toxicity: A Defense Mediated by Sulphur-Rich Molecules and Oxidative Prevention Systems"

_ijms, 2025, doi:10.3390/ijms262010179_

Round 1
Reviewer 1 Report
Comments and Suggestions for Authors
Review of manuscript ijms-3852128 "Bacillus subtilis response to mercury toxicity: A defense mediated by sulphur-rich molecules and oxidative prevention systems"
This study has assessed the impact of mercury exposure on Bacillus subtilis cells in several ways, including analysis of survival of wildtype and selected mutants, mutation frequencies and sequence change tendencies, as well as global changes in gene expression. The experiments and conclusions appear to be largely reasonable but I would request that the authors address several concerns.
Major items:
- Regarding the data on expression levels in Tables S1 and S2, it would be helpful to describe the cutoff used to select which genes were expressed at different levels by how many fold higher or lower, e.g., > 2-fold or >3-fold, etc., than untreated cells, even if it is only mentioned once. Software-based cutoffs are often difficult to interpret.
- In Figure S1 it difficult to understand the differences among the different mutant strains when they are in different graphs with different y-axis scales. It would be more understandable if WT 168 and the mutants were put on the same graph using a vertically tall graph to separate the lines as much as possible.
- The very detailed model in Figure 4 is somewhat speculative based on the modest amount of data obtained in the current study. There is an extensive literature on mercury effects, especially in eukaryotes, and only a small part of it is discussed. It would be helpful to discuss more of this literature when describing Figure 4 and explain how its branches are supported, or not supported, by past results on interaction of Hg with thiols, iron metabolism, ROS generation and intracellular metabolic pathways.
- At the end of the Abstract the authors state "We hypothesize that free iron...". This manner of expression may be okay, but it is more normal to summarize what the data from the current study suggests rather than to end with a hypothesis, e.g., "The aggregate data suggests that ..." or "These results and other data suggest that...".
Minor items:
- Line 96, change "and Rifampicin" to "or Rifampicin"
- Line 153, change read reads to reads
- Line 171, the text says "standard error (SD)" but SD means standard deviation. Here and in the legend of Figure S1, was SE or SD used?
- Line 190, "the strain deficient for SodA exhibited higher lethal doses 90/50´s to 190
mercury treatment that the wild-type strain (Table 2)." --> it looks like the LD50 of sodA is not higher than WT, so should the sentence refer only to LD90?
- Line 209, "The asterisks indicate statistical differences (T student P < 0.05) with respect to the WT strain. The double asterisk indicates a statistical difference between the treatment or not with HgCl2 (T student P <0.05)." --> does the double asterisk actually mean p < 0.01?
Author Response
"Please see the attachment." (Responses to Reviewer 1)

Reviewer 2 Report
Comments and Suggestions for Authors
This is a comprehensive study on the effect of mercury ions on different prokaryotic responses through the lens of Bacillus subtilis. The authors have determined the LD50 and LD90 followed by mutagenic effects and gene expression profiles in response to mercury. Statistical analyses have been provided where necessary. At the end of the manuscript, a model is provided to explain the effects of mercury on the bacterial cell.
Some improvements that are recommended are:
- It will be helpful to the reader to provide the results as a graph that supports the table 2 that has been provided.
- It would be helpful to provide a rationale on why it is important to study the effects of mercury in Bacillus subtilis.
The overall English is fine but there are minor grammatical errors that can be corrected.
Author Response
"Please see the attachment." (Response to Reviewer 2)

Reviewer 3 Report
Comments and Suggestions for Authors
The aim of the proposed was to evaluate the mutagenic potential of Hg (II) ions and prevention/repair systems in counteracting the genetic damage inflicted by mercury ions in B. subtilis. The findings of proposed objectives showed that thiol-rich molecules block the oxidative damage and toxic effects induced by Hg (II) ions. It is a good piece of research works to understand the molecular mechanism of defense system in microorgamism. However, the submitted manuscript requires major revision and manuscript in its current form is not acceptable for publication in the esteemed “International Journal of Molecular Sciences” journal. Authors should ensure that they have followed the Instructions to Authors of this journal. It is requested that authors must revise the manuscript according to suggested corrections. The corrections made in the manuscript should be highlighted. So, it would be easier to identify the modified content from the original submitted manuscript.
Comments:
- Please revise the abstract in subheads like background, objectives, methods, findings and important conclusions to make it more impactful and clear.
- Abstract and introduction sections lack clear objectives.
- What was the basis for selection of subtilis for this designed study? Please mention in the introduction section.
- Manuscript has high plagiarism content which should not be more than 10%.
- In Statistical analysis section, T student test should be Student's t-test and P value should be p value (p <0.05).
- It will be more impactful if authors will add short information related to mutant like mutated gene/genes and its product roles in cellular defense system in Materials and Methods section.
- Please verify the data of Wild type 168 and ∆sodA mentioned in Table 2. SOD plays very important role in cellular defense system and both wild and mutant have almost similar LD50 value.
- Please check the data presented in the Table 3 and Figure 1. Both data are unmatched.
- Figure 1, p <0.05 are for * or **. Please clearly mention the level of significance p <0.05; p <0.01; p <0.001 in Figure legend.
- The name of gene should be uniform throughout manuscript.
- On Page no. 12; Please verify the statement: Notably, the lack of SodA, an antioxidant enzyme that converts superoxide radicals into hydrogen peroxide [35] increased B. subtilis resistance to mercury ions, above that of the wild-type strain However in Table 2 LD50 are 1.19 ± 0.04 for wild type and 1.18 ± 0.09 for ∆sodA.
- Conclusion section is missing. Please add this section.
Author Response
"Please see the attachment." (Response to reviewer 3)

Round 2
Reviewer 3 Report
Comments and Suggestions for Authors
The revised manuscript can be accepted for publication in esteemed Journal "International Journal of Molecular Sciences". Authors have addressed each queries very well raised by reviewer and have critically modify the manuscript as per requirement.
Author Response
We want to thank the reviewer for their critical and constructive comments to our manuscript. Undoubtedly, this manuscript was substantially improved.
kind regards